# Emotional Competencies and Psychological Well-Being in Costa Rican Emerging Adults: The Mediating Role of Self-Esteem and Resilience

**DOI:** 10.3390/ejihpe15050089

**Published:** 2025-05-20

**Authors:** María Teresa Dobles Villegas, Hugo Sanchez-Sanchez, Konstanze Schoeps, Inmaculada Montoya-Castilla

**Affiliations:** Department of Personality, Assessment and Psychological Treatment, University of Valencia, Av. Blasco Ibáñez 21, 46010 Valencia, Spain; matedovi@alumni.uv.es (M.T.D.V.); hugo.a.sanchez@uv.es (H.S.-S.); inmaculada.montoya@uv.es (I.M.-C.)

**Keywords:** psychological well-being, emotional competencies, self-esteem, resilience, emerging adulthood, vulnerable population, Caribbean population, emotion management and regulation, positive relationships, environmental mastery

## Abstract

There is strong empirical evidence on the benefits of emotional competencies, self-esteem, and resilience for well-being in the youth and adult populations. However, little research has been conducted to identify protective factors for well-being among emerging adults in rural areas of Costa Rica, which are particularly vulnerable. This study aims to examine the relationships between emotional competencies and dimensions of psychological well-being, as well as the mediating role of self-esteem and resilience in Costa Rican university students. The sample consisted of 328 students aged 18 to 30 years (*M* = 21.31, *SD* = 3.28), of whom 47.90% were women. This study utilized Ryff’s Psychological Well-being Scale, the Emotional Skills and Competence Questionnaire, Rosenberg’s General Self-esteem Scale, and the Connor–Davidson Resilience Scale. The results indicated that emotional competencies, particularly emotion management and regulation, have direct positive effects on psychological well-being. Additionally, self-esteem played a mediating role, showing indirect effects between emotional competencies and the dimensions of psychological well-being. Resilience had a less pronounced mediating role than self-esteem in terms of effect size and the number of significant relationships. Moreover, negative effects were identified between emotional perception and understanding and certain well-being dimensions, such as positive relationships and environmental mastery. The findings provide evidence that emotional competencies, self-esteem, and resilience are key factors in promoting psychological well-being among emerging adults in rural areas of Costa Rica. These results highlight the importance of fostering emotional skills and strengthening self-esteem, particularly in emerging adults from socioeconomically disadvantaged backgrounds.

## 1. Introduction

The growing interest in promoting emotional competencies has gained increasing relevance in scientific research, particularly in relation to psychological well-being ([6]; [46]). This interest has been further reinforced by the impact of the COVID-19 pandemic worldwide ([27]; [63]), positioning well-being promotion as a priority in research and mental health policies ([52]).

From a positive psychology perspective, the Well-being Promotion Model provides a relevant conceptual framework for understanding the impact of emotional competencies on mental health and well-being promotion ([23]). This model defines well-being as the result of achieving self-fulfillment and leading a meaningful life, contrasting with approaches focused solely on illness. Moreover, this theory has become a reference point across various fields of research ([25]; [50], [51]). This practical model focuses on improving well-being in real-world contexts, such as healthcare, education, and the community. Although it often incorporates Ryff’s theory of psychological well-being as a basis, it extends it by connecting it to Neuman’s systems theory, which views people holistically and in constant interaction with their environment ([23]). Furthermore, it emphasizes emotional competencies as key tools for fostering well-being, offering concrete strategies and interventions that actively promote well-being ([23]; [50], [51]).

As an applied framework, the model conceptualizes protective factors of well-being as psychological resources that are activated in response to stressors with the aim of shielding individuals from negative impacts. Among these protective factors, self-esteem and resilience are highlighted for their significant role in supporting well-being ([23]; [25]).

Ryff’s Psychological Well-being Theory is articulated through six interrelated dimensions. Self-acceptance is defined by the integration of personal strengths and weaknesses; Autonomy is defined as the capacity for independent action in accordance with one’s own values; Mastery of the environment is characterized by the effective management of external demands and the creation of favorable contexts; Personal growth is defined as a sense of continuous development and openness to new experiences; Positive relationships with others emphasize the importance of meaningful bonds based on empathy and trust; and Purpose in life is defined as the existence of goals and meaning in life ([50]; [53]).

To complement this central framework, Mayer and Salovey’s model of emotional intelligence provides a valuable theoretical foundation for understanding the emotional abilities involved in psychological well-being ([38]). Their model conceptualizes emotional competencies as a set of skills. These include the perception and understanding of emotions, the ability to perceive, identify and appropriately express one’s own and others’ emotions and understand their meaning, the expression of emotions, the ability to appropriately communicate emotional states and the management and regulation of emotions, and the ability to manage and modulate emotions constructively both in oneself and in interaction with others ([11]; [13]).

Numerous studies have examined the relationship between psychological well-being and emotional competencies, consistently finding a positive association between these constructs ([10]; [20]; [55]). Specifically, high scores in emotional understanding and regulation have been identified as strong predictors of all dimensions of psychological well-being ([7]). Similarly, emotional perception has been linked to well-being outcomes ([56]). However, some studies have reported negative correlations with the autonomy dimension, while positive direct effects have been observed in areas such as purpose in life and personal growth ([8]).

Beyond emotional competencies, self-esteem has emerged as a key variable in this relationship, mediating between emotional skills and psychological well-being. Research suggests that greater abilities in perceiving, understanding, and regulating emotions contribute to a more positive self-evaluation, which in turn enhances psychological well-being ([34]).

Self-esteem is defined as an individual’s positive or negative evaluation of oneself, the acceptance or rejection of one’s personal development, and the belief in being sufficiently capable ([47]; [48]). In emerging adulthood, self-esteem has been identified as a protective factor, as higher levels of self-esteem predict success and well-being in areas such as social relationships, employment, and mental health ([41]).

Furthermore, various studies have shown that self-esteem mediates different relationships, including those between parenting styles and depressive and anxious symptoms ([21]), life satisfaction and social support ([29]) parental attachment and loneliness in adolescents ([31]), and insecure attachment and subjective well-being ([32]). Additionally, self-esteem has been found to mediate the relationship between emotional competencies and internalizing symptoms ([54]), as well as between emotional competencies and perceived stress ([62]).

Along with self-esteem, resilience is another central variable in this study due to its strong connection to psychological well-being and its relevance to the target population ([60]). Resilience is defined as the ability to maintain or enhance psychological stability in the face of adversity ([17]) and has been crucial in understanding how some emerging adults exposed to risk factors manage to avoid developing psychopathologies ([26]; [40]).

Resilience plays a fundamental role during this period, facilitating adaptation to stress and change ([26]; [40]). Studies have also demonstrated that resilience acts as a mediator in various relationships. For instance, it has been identified as a mediator between adverse childhood experiences and Internet gaming disorder in university students ([35]). Additionally, research has explored resilience in conjunction with self-esteem, leading to the concept of “ego-resilience”, which has been described as a mediator between perceived success in physical education and school adaptation, with the idea that people with high self-esteem are more likely to believe in their ability to cope with and overcome challenges ([16]).

Furthermore, resilience and social support have been found to be significant mediators in the relationship between emotional competencies and psychological well-being in post-university emerging adults ([57]).

Since psychological well-being is influenced by multiple factors, including emotional competencies, self-esteem, and resilience, it is essential to analyze these dynamics within the context of emerging adulthood ([1]). This stage of human development, which spans approximately from ages 18 to 29, is characterized as a period of identity exploration, instability, and transition into full adulthood ([1]). It is also marked by prolonged education and training, during which young people explore various life possibilities ([3]). Emerging adulthood is a crucial developmental stage that has garnered significant attention in recent decades due to profound social, cultural, and economic changes ([5]).

The trajectory of this period can vary depending on social and cultural contexts ([2]). In industrial or post-industrial settings with greater access to higher education, individuals navigate distinct pathways in their careers, academic training, and housing. Conversely, in other contexts, such as developing countries, emerging adults face significantly different challenges ([4]; [58]).

In Costa Rica, these transitions are shaped by macrostructural factors such as economic inequality, the education crisis, multidimensional poverty, and difficulties in labor market integration ([18]). These challenges were exacerbated by the pandemic, with a greater impact on rural areas ([39]). Such conditions can amplify the uncertainty and stress associated with emerging adulthood ([26]), underscoring the importance of examining how variables such as emotional competencies, self-esteem, and resilience influence psychological well-being during this developmental stage.

Despite the growing body of research on emotional competencies and well-being, few studies have examined these relationships in the context of emerging adulthood in Latin America, particularly in Costa Rica, where socioeconomic challenges may uniquely shape these dynamics. By examining these relationships, this study aims to provide valuable insights into how emotional competencies, self-esteem, and resilience can be leveraged to promote psychological well-being in emerging adults, particularly in socioeconomically disadvantaged contexts. To this end, the following hypotheses are proposed: H1: Emotional competencies will be positively associated with the dimensions of psychological well-being. H2: Self-esteem will be positively associated with both emotional competencies and the dimensions of psychological well-being. H3: Resilience will be positively associated with both emotional competencies and the dimensions of psychological well-being. H4: Self-esteem will mediate the relationship between emotional competencies and the dimensions of psychological well-being. H5: Resilience will mediate the relationship between emotional competencies and the dimensions of psychological well-being.

## 2. Method

### 2.1. Participants

Our study sample consisted of 328 Costa Rican university students enrolled at the Universidad Nacional, Sarapiquí Campus, in 2024. Of the participants, 47.9% were women, with ages ranging from 18 to 30 years (*M* = 21.31, *SD* = 3.28). Regarding their fields of study, 54.7% were enrolled in programs related to Administration and Commerce, 30.5% in Information Technology, 10.7% in English, and 4.3% in Education.

The sample size was determined using a 95% confidence level and a maximum acceptable margin of error of 5%. Considering that Costa Rica’s five public universities serve a total student population of 119,108 ([44]). The expected sample size was *n* = 383 students. Inclusion criteria required participants to be between 18 and 30 years old and to provide informed consent for data use, leading to the exclusion of five individuals.

According to indicators from Costa Rica’s Ministry of National Planning and Economic Policy, most participants (62.2%) came from areas classified as having a very low social development index, while 4% were from high social development areas. The remaining 33.8% resided in regions with medium to low social development indices. Additionally, 68.05% of participants reported that their parents had an educational level below incomplete secondary education, 23.75% had parents with higher education, and 7.9% either did not know or preferred not to answer.

### 2.2. Instruments

Psychological well-being was assessed using Ryff’s Psychological Well-Being Scale ([49]), specifically the version validated in the Mexican population ([42]). This self-report scale consists of 18 items (e.g., “I feel like I have grown a lot as a person recently”), rated on a Likert scale from 1 (strongly disagree) to 6 (strongly agree) and evaluates six dimensions, as follows: self-acceptance, positive relationships, autonomy, environmental mastery, personal growth, and purpose in life. The reliability coefficients for each dimension are as follows: self-acceptance (α = 0.76; ω = 0.61); positive relationships (α = 0.67; ω = 0.44); autonomy (α = 0.59; ω = 0.59); environmental mastery (α = 0.56; ω = 0.51); personal growth (α = 0.69; ω = 0.51); and purpose in life (α = 0.78; ω = 0.75).

Emotional competencies were assessed using the Emotional Skills and Competence Questionnaire (ESCQ) ([59]), with the Mexican adaptation by [61] ([61]). This brief version consists of 20 items (e.g., “I am good at expressing how I feel”), rated on a Likert scale from 1 (never) to 5 (always), and evaluates three factors, as follows: perceiving and understanding emotions, expressing emotions, and managing and regulating emotions. The following reliability coefficients are acceptable: perceiving and understanding emotions (α = 0.75); managing and regulating emotions (α = 0.76); and expressing and naming emotions (α = 0.90).

Self-esteem was measured using the Rosenberg Self-Esteem Scale (RSES) ([37]), which assesses the perception of being “good enough”. The scale consists of 10 items (e.g., “I have a positive opinion of myself”), rated on a Likert scale from 1 (strongly disagree) to 4 (strongly agree). It has demonstrated strong validity and consistency, supported by extensive empirical evidence over time ([15]; [37]). The version used ([15]) reported an internal consistency coefficient (Cronbach’s alpha) of α = 0.80.

Resilience was assessed using the Connor–Davidson Resilience Scale (CD-RISC), specifically the short Spanish version known as CD-RISC 10 ([17]). This unidimensional scale consists of 10 items (e.g., “I achieve my goals despite difficulties”), rated on a Likert scale from 0 (not at all) to 5 (almost always). The scale has a reported reliability of α = 0.85.

### 2.3. Procedure

To conduct this study, approval was obtained from the Ethics Committee of the National University of Costa Rica (UNA-CECUNA-OFI-036-2024), with the project registered as UNA-CECUNA-2024-P001. This approval ensured compliance with ethical standards, informed consent, and data protection for participants. The survey was administered online through the LimeSurvey platform, with an average completion time of approximately 35 min. Prior to participation, students provided informed consent, as approved by the Ethics Committee of the National University of Costa Rica. Participant recruitment was conducted in classrooms with the support of course instructors, who assisted in distributing the survey.

### 2.4. Statistical Analyses

Data analyses were conducted using SPSS v.25 and Mplus v.8. Descriptive statistics, including mean, standard deviation, skewness, kurtosis, minimum and maximum scores, reliability indicators (Cronbach’s alpha, McDonald’s Omega) were examined for all studied variables. In addition, Pearson’s bivariate correlations between variables were calculated. Furthermore, we conducted mediation analysis examining direct and indirect effects using maximum likelihood estimation, following the model fit indices recommended by Hu and Bentler: χ^2^: (Chi-square), Root Mean Square Error of Approximation (RMSEA), Comparative Fit Index (CFI), Tucker–Lewis Index (TLI), Standardized Root Mean Square Residual (SRMR), and *R*^2^ (Coefficient of Determination) ([12]; [28]). Additionally, the analyses adhered to the AGREMA guidelines, a checklist designed to provide recommendations for studies reporting statistical mediation analyses ([14]; [33]).

## 3. Results

### 3.1. Descriptive and Correlational Statistics

Descriptive analyses provide the mean, standard deviation, and other statistical indicators that characterize our study data (see Table 1).

Table 2 presents the correlation matrix of the analyzed variables. Notably, a medium-to-high and significant correlation was observed between self-esteem and resilience. Self-esteem correlated positively with all variables, showing the highest correlation with self-acceptance and the lowest with perceiving and understanding emotions. Additionally, resilience exhibited a similar pattern, with its strongest correlation found with emotional regulation and management, while the lowest correlation was observed with positive relationships.

### 3.2. Mediation Analysis

An initial model was tested, considering all direct and indirect effects among the variables of interest. The fit analysis for this initial model revealed poor fit indices, including a chi-square value outside the acceptable range, CFI = 0.84, RMSEA = 0.17, SRMR = 0.09, and TLI = 0.77.

These results indicate a substantial model misfit ([14]), likely due to the high number of variables included, leading to more than 30 estimated relationships.

To improve model fit, a revised model was tested by removing non-significant paths and allowing for covariation between self-esteem and resilience (*r* = 0.61), suggesting that these constructs share a substantial proportion of variance. This refined model demonstrated satisfactory fit indices, χ^2^ = 19.44 with 16 degrees of freedom (*p* = 0.24), RMSEA = 0.03, CFI = 0.99, TLI = 0.99, and SRMR = 0.02 ([12]). Additionally, the final model was statistically significant and explained a substantial percentage of variance, reaching approximately 40% for some dependent variables (see Table 3).

### 3.3. Direct Effects

The results of the direct effects in the final model (see Table 4) indicate that emotional regulation and management, as well as emotional expression, had significant and positive direct effects on multiple dimensions of well-being. Notably, emotional expression had strong positive effects on purpose in life, personal growth, and self-acceptance, while emotional regulation and management were positively associated with self-acceptance and personal growth.

Conversely, perceiving and understanding emotions exhibited a strong negative direct effect on positive relationships, marking the most substantial negative effect among the analyzed variables. Additionally, this dimension also had negative direct effects on self-acceptance and environmental mastery, while showing a positive effect on personal growth.

Furthermore, no emotional competency showed a direct effect on the autonomy dimension of psychological well-being, whereas all other dimensions of well-being had at least one significant direct relationship. Regarding the direct effects of the mediating variables, perceiving and understanding emotions had a significant negative effect on self-esteem, while all other direct relationships with the remaining variables were positive (see Figure 1).

### 3.4. Indirect Effects

The analysis of indirect effects in the final model revealed that self-esteem significantly mediated all relationships between the three emotional competencies and the six dimensions of psychological well-being (see Table 5). Self-esteem exhibited both positive and negative indirect effects in these relationships. Specifically, self-esteem negatively mediated the relationship between perceiving and understanding emotions and all dimensions of psychological well-being. In contrast, it positively mediated the relationship between emotional regulation and management, as well as emotional expression, with all dimensions of psychological well-being. Notably, the strongest mediation effect of self-esteem was observed between emotional expression and autonomy, followed by environmental mastery and self-acceptance.

In comparison, resilience exhibited fewer mediation effects than self-esteem. The estimates for resilience were close to zero and non-significant for the relationship between perceiving and understanding emotions and all dimensions of psychological well-being. The indirect effects of resilience were significant for emotional regulation and management, as well as for emotional expression, though with small effect sizes across all dimensions of well-being, except for personal growth and positive relationships, where they were not significant.

## 4. Discussion

This study highlights the importance of considering the interactions among variables that influence psychological well-being, such as emotional competencies, self-esteem, and resilience, in the Costa Rican emerging adult population. Investigating these relationships provides a deeper understanding of the processes that promote adjustment and mental health in this vulnerable population. The findings offer a comprehensive perspective on the relationships between emotional competencies, self-esteem, resilience, and psychological well-being among Costa Rican emerging adults.

The results indicate that emotional competencies are associated with psychological well-being, though this relationship is not uniform. On the one hand, the dimensions of emotional regulation and management, as well as emotional expression, demonstrated significant positive effects on key dimensions of well-being, such as purpose in life, personal growth, and self-acceptance, consistent with previous findings ([10]; [20]; [36]; [55]).

In contrast, the dimension of perceiving and understanding emotions had negative effects on positive relationships, environmental mastery, and self-acceptance, similar to findings in other studies ([8]), where negative associations were primarily observed in the autonomy dimension of psychological well-being. This may suggest that an excessive focus on perceiving and processing emotions heightens sensitivity to negative aspects of the environment, potentially undermining certain aspects of psychological well-being.

The hypothesis proposing a positive relationship between self-esteem, emotional competencies, and psychological well-being, with self-esteem as a mediator, was supported by the findings. These results align with the existing literature ([19]; [24]; [34]), suggesting that during emerging adulthood, self-esteem acts as a protective factor. Feeling valued and capable facilitates more adaptive emotional management, helping individuals navigate the challenges characteristic of this developmental stage ([41]).

Accordingly, the development of strong emotional competencies may improve stress and uncertainty management—key elements of this period ([62]), contributing to higher levels of psychological well-being ([34]). This effect may be particularly relevant in contexts such as Costa Rica, where socioeconomic and cultural challenges intensify the need for protective factors to support the transition to adulthood ([26]).

The mediation analysis revealed that self-esteem acts as a significant mediator in most of the relationships analyzed, with a particularly strong positive impact on emotional management, regulation, and expression ([9]). This suggests that individuals with higher self-esteem may feel more confident when facing emotionally challenging situations, enabling them to express and regulate their emotions more effectively. This, in turn, translates into higher levels of self-acceptance, purpose in life, personal growth, and other dimensions of psychological well-being ([19]).

The negative mediation between perceiving and understanding emotions, self-esteem, and psychological well-being suggests that in some cases, high self-esteem may contribute to biases in emotional interpretation ([22]). Individuals with elevated self-esteem may overestimate their ability to understand emotions, leading to a distorted perception of their emotional environment and a subsequent decline in psychological well-being. Similarly, self-esteem driven by self-imposed standards could encourage self-critical and ruminative emotional processing, negatively impacting well-being ([30]). Likewise, heightened emotional perception without sufficient confidence in personal coping resources may generate insecurity, stress, and discomfort. These findings underscore the importance of balancing emotional awareness with a healthy level of self-esteem to foster psychological well-being ([45]).

Regarding resilience, its relationship with emotional competencies and psychological well-being, as well as its mediating role, aligns with prior research ([34]; [43]; [57]; [60]). This is particularly relevant in contexts with resource limitations and a high prevalence of risk factors, where higher resilience levels may facilitate effective emotional management. Resilience equips individuals with tools to navigate both environmental challenges and the demands of emerging adulthood ([26]), potentially contributing to higher levels of psychological well-being. The development of stronger emotional competencies fosters better adaptation to the demands and transitions of this life stage ([40]; [60]).

Similarly, resilience demonstrated a considerable mediating effect alongside self-esteem, consistent with the concept of “ego-resilience” proposed in other studies ([16]). Resilience provides individuals with the resources needed to adapt to adversity and stress, while self-esteem offers a positive self-evaluation that enhances emotional management and well-being ([24]; [34]). Thus, the combination of these two protective factors may strengthen individuals’ capacity to regulate their emotions and cope with the challenges of emerging adulthood, reducing their vulnerability to psychological distress in the face of difficulties typical of this developmental stage in Costa Rica ([26]; [40]; [60]).

Based on these findings, the increasing social, emotional, and psychological demands faced by Costa Rican university students underscore the need for integrating emotional development programs into higher education. Such programs should emphasize the development of emotional competencies, the promotion of psychological well-being, and the reinforcement of self-esteem and resilience to foster overall mental health ([57]). Future research should explore the influence of socioeconomic factors on resilience and their interaction with emotional competencies, assess interventions that strengthen emotional competencies among low-income university students, and evaluate their impact on mental health. Additionally, longitudinal studies examining changes in psychological well-being, self-esteem, and emotional competencies over time would enhance the understanding of well-being in this population and inform the design of more effective support policies and programs.

Finally, certain limitations should be considered when interpreting the results of this study. First, its cross-sectional design prevents the establishment of causal relationships between the analyzed variables, highlighting the need for future longitudinal studies to provide a more in-depth understanding of how psychological well-being and emotional competencies evolve over time. Additionally, self-report measures may be subject to social desirability biases or participants’ subjective interpretations. In addition, the questionnaires leave out complex and deep emotional elements from the subject’s response, so future studies should add other methodological strategies to account for these aspects.

Moreover, the present study utilized total scale scores for its analyses; however, item-level analyses have been posited as a means of enhancing the discriminant validity of the scale. It is recommended that future studies adopt such approaches to explore latent structures and to control for possible covariance between variables. Despite these limitations, this study offers valuable insights into the role of emotional competencies, self-esteem, and resilience in psychological well-being, laying the groundwork for future research that employs more robust methodologies and more diverse samples.

## 5. Conclusions

This study concludes that emotional competencies, self-esteem, and resilience are key determinants of psychological well-being among Costa Rican university students during the critical period of emerging adulthood. The positive relationship between emotional expression, self-esteem, and key dimensions of well-being, such as purpose in life and personal growth, underscores the need to incorporate educational and community strategies that prioritize the development of emotional competencies.

The negative indirect effects between emotional perception and understanding, self-esteem, and psychological well-being highlight the importance of balancing emotional perception and understanding with self-esteem to promote well-being. In this regard, it is essential to recognize that a high capacity to perceive and understand emotions is not sufficient on its own; this ability must be accompanied by healthy self-esteem to interpret emotional information appropriately.

## Figures and Tables

**Figure 1 ejihpe-15-00089-f001:**
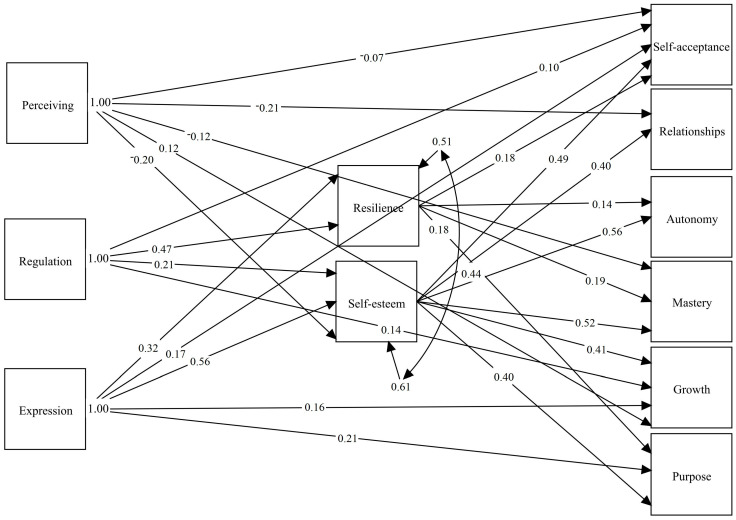
Mediation Diagram. *Note*: Perceiving = Perceiving and understanding emotions; Regulation = Emotional regulation and management; Expression = Emotional expression; Resilience = Resilience; Self-esteem = Self-esteem; Self-acceptance = Self-acceptance; Relationships = Positive relationships with others; Autonomy = Autonomy; Mastery = Environmental mastery; Growth = Personal growth; Purpose = Purpose in life.

**Table 1 ejihpe-15-00089-t001:** Descriptive Statistics for the Variables.

Variable	*M*	*SD*	*Skew*	*Kurt*	*Min*	*Max*	α	ω
Perceiving and understanding emotions	2.89	0.57	−0.36	1.10	1	4	0.87	0.87
Emotional regulation and management	2.95	0.63	−0.48	0.15	1	4	0.75	0.75
Emotional expression	2.80	0.64	−0.25	−0.06	1	4	0.89	0.89
Self-esteem	28.67	6.51	−0.33	−0.09	10	40	0.89	0.88
Resilience	24.80	7.30	−0.48	0.82	0	40	0.87	0.87
Self-acceptance	4.17	1.28	−0.60	−0.32	1	6	0.86	0.86
Positive relationships	3.76	1.30	−0.09	−0.59	1	6	0.69	0.71
Autonomy	4.15	1.24	−0.35	−0.59	1	6	0.65	0.73
Environmental mastery	3.64	1.17	0.17	−0.60	1	6	0.61	0.66
Purpose in life	4.86	1.22	−1.42	1.75	1	6	0.92	0.88
Personal growth	4.24	1.23	−0.70	−0.13	1	6	0.92	0.92

*Note*: *M* = Mean; *SD* = Standard Deviation; *Skew* = Skewness; *Kurt* = Kurtosis; *Min* = Minimum score; *Max* = Maximum score; α = Cronbach’s alpha; ω = McDonald’s Omega.

**Table 2 ejihpe-15-00089-t002:** Correlation Matrix of this Study.

Variables	1	2	3	4	5	6	7	8	9	10
1. Self-esteem	—									
2. Resilience	0.64 ***	—								
3. Perceiving emotions	0.19 ***	0.39 ***	—							
4. Emotional regulation	0.46 ***	0.66 ***	0.41 ***	—						
5. Emotional expression	0.58 ***	0.59 ***	0.52 ***	0.59 ***	—					
6. Self-acceptance	0.73 ***	0.65 ***	0.24 ***	0.53 ***	0.59 ***	—				
7. Positive relationships	0.36 ***	0.18 **	−0.13 *	0.12 *	0.18 **	0.18 **	—			
8. Autonomy	0.65 ***	0.50 ***	0.16 **	0.31 ***	0.43 ***	0.54 ***	0.42 ***	—		
9. Environmental mastery	0.61 ***	0.48 ***	0.05	0.34 ***	0.42 ***	0.56 ***	0.48 ***	0.51 ***	—	
10. Purpose in life	0.58 ***	0.55 ***	0.35 ***	0.48 ***	0.54 ***	0.74 ***	0.00	0.36 ***	0.39 ***	—
11. Personal growth	0.64 ***	0.59 ***	0.28 ***	0.46 ***	0.56 ***	0.83 ***	0.12 *	0.45 ***	0.55 ***	0.74 ***

*Note*: * *p* < 0.05, ** *p* < 0.01, *** *p* < 0.001.

**Table 3 ejihpe-15-00089-t003:** Coefficient of Determination.

Variables	*R* ^2^	Standard Error	*p*
Self-acceptance	0.61	0.04	<0.001
Positive relationships	0.17	0.04	<0.001
Autonomy	0.44	0.05	<0.001
Environmental mastery	0.41	0.04	<0.001
Purpose in life	0.48	0.05	<0.001
Personal growth	0.43	0.05	<0.001
Self-esteem	0.39	0.5	<0.001
Resilience	0.49	0.4	<0.001

*Note*: *R*^2^ = Variance explained by the independent variable; *p* = *p*-value of significance.

**Table 4 ejihpe-15-00089-t004:** Direct Effects Final Model.

IV	DV	*Est*	*SE*	LL 2.5%	UL2.5%
Perceiving emotions	Self-acceptance	−0.07 *	0.03	−0.12	−0.01
	Positive relationships	−0.21 ***	0.06	−0.32	−0.10
	Autonomy	-	-	-	-
	Environmental mastery	−0.13 **	0.05	−0.22	−0.03
	Personal growth	0.12 *	0.05	0.02	0.23
	Purpose in life	-	-	-	-
Emotional regulation	Self-acceptance	0.11 *	0.05	0.01	0.20
	Positive relationships	-	-	-	-
	Autonomy	-	-	-	-
	Environmental mastery	-	-	-	-
	Personal growth	0.14 **	0.05	0.04	0.23
	Purpose in life	-	-	-	-
Emotional expression	Self-acceptance	0.17 **	0.05	0.07	0.27
	Positive relationships	-	-	-	-
	Autonomy	-	-	-	-
	Environmental mastery	-	-	-	-
	Personal growth	0.16 *	0.06	0.04	0.28
	Purpose in life	0.22 ***	0.06	0.11	0.32

*Note*: * *p* < 0.05, ** *p* < 0.01, *** *p* < 0.001; IV = Independent Variable; DV = Dependent Variable; *Est* = Estimate; *SE* = Standard Error; LL = Lower Limit of 95% Confidence Interval; UL = Upper Limit of 95% Confidence Interval.

**Table 5 ejihpe-15-00089-t005:** Indirect Effects Final Model.

IV	*M*	DV	*Est*	*SE*	LL 2.5%	UL2.5%
Perceiving emotions	Self-esteem	Self-acceptance	−0.10 ***	0.03	−0.15	−0.05
		Positive relationships	−0.08 **	0.03	−0.13	−0.03
		Autonomy	−0.11 ***	0.03	−0.18	−0.05
		Environmental mastery	−0.10 **	0.03	−0.16	−0.05
		Personal growth	−0.08 ***	0.02	−0.12	−0.04
		Purpose in life	−0.08 ***	0.02	−0.12	−0.04
Emotional regulation	Resilience	Self-acceptance	0.08 **	0.03	0.02	0.14
	Self-esteem		0.10 **	0.03	0.04	0.16
	Self-esteem	Positive relationships	0.08 **	0.03	0.03	0.14
	Resilience	Autonomy	0.07 *	0.03	0.02	0.12
	Self-esteem		0.12 **	0.03	0.05	0.18
	Resilience	Environmental mastery	0.09 **	0.03	0.04	0.14
	Self-esteem		0.11 **	0.03	0.05	0.17
	Self-esteem	Personal growth	0.09 **	0.03	0.03	0.14
	Resilience	Purpose in life	0.09 **	0.03	0.03	0.15
	Self-esteem		0.08 **	0.03	0.03	0.13
Emotional expression	Resilience	Self-acceptance	0.06 **	0.02	0.02	0.10
	Self-esteem		0.28 ***	0.04	0.21	0.35
	Self-esteem	Positive relationships	0.23 ***	0.04	0.15	0.31
	Resilience	Autonomy	0.04 **	0.02	0.01	0.08
	Self-esteem		0.32 ***	0.05	0.22	0.41
	Resilience	Environmental mastery	0.06 ***	0.02	0.03	0.09
	Self-esteem		0.29 ***	0.05	0.20	0.39
	Self-esteem	Personal growth	0.23 ***	0.04	0.15	0.31
	Resilience	Purpose in life	0.06 **	0.02	0.02	0.10
	Self-esteem		0.23 ***	0.03	0.16	0.29

*Note*: * *p* < 0.05, ** *p* < 0.01, *** *p* < 0.001; IV = Independent Variable; *M* = Mean; DV = Dependent Variable; *Est* = Estimate; *SE* = Standard Error; LL = Lower Limit of 95% Confidence Interval; UL = Upper Limit of 95% Confidence Interval.

## Data Availability

The raw data supporting the conclusions of this article will be made available by the authors on request.

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
