# Peer review of "Emotional Competencies and Psychological Well-Being in Costa Rican Emerging Adults: The Mediating Role of Self-Esteem and Resilience"

_ejihpe, 2025, doi:10.3390/ejihpe15050089_

Round 1
Reviewer 1 Report
Comments and Suggestions for Authors
Please find the comments for your investigation. Overall, the paper is prepared well. I would suggest demonstrating theoretical and empirical distinction between the studied constructs to avoid jingle and jangle fallacies.
Hanfstingl B, Oberleiter S, Pietschnig J, Tran US and Voracek M (2024) Detecting jingle and jangle fallacies by identifying consistencies and variabilities in study specifications – a call for research. Front. Psychol. 15:1404060. doi: 10.3389/fpsyg.2024.1404060
Altgassen, E., Geiger, M., & Wilhelm, O. (2023). Do you mind a closer look? A jingle-jangle fallacy perspective on mindfulness. European Journal of Personality, 38(2), 365-387. https://doi.org/10.1177/08902070231174575 (Original work published 2024)
- Please add more keywords (up to 10), it helps with indexation.
- the introduction describes several theoretical models, however, a common theoretical model should should be created. Please elaborate on the connection between the variables. Overall, if you study personal resources and well-being as a main variable, potential theoretical models related to resources and well-being should be considered.
- Reliability coefficients in this sample can be reported in Table 1 as these data are part of results.
- Please indicate examples of statements used in each subscale in the instruments.
- When describing statistical analyses, I would suggest be more specific and indicate for instance the type of correlation (e.g., Pearson, Spearman etc.). This information is not provided.
- Please format tables for better readability. Please decrease the font size.
- Fit indices should be described in statistical analysis section. There are issues with the abbreviation use. Please make sure that abbreviations are deciphered when introducing.
- P-values cannot equal 0.000, in such cases use p < 0.001.
- It seems that covariance between self-esteem and resilience was introduced (Figure 1). Please confirm and elaborate on this if it is the case.
- Please elaborate on potential discriminant validity issues of the studied variables. Theoretical and empirical distinction of these constructs should be demonstrated, especially between self-esteem and well-being.
Author Response
Comment 1: Please add more keywords (up to 10), it helps with indexation.
Response 1: Additional keywords have been incorporated to improve indexation: “Caribbean Population; Emotion management and regulation; Positive relationships; Environmental mastery” (see lines 38–39).
Comment 2: The introduction describes several theoretical models, however, a common theoretical model should be created. Please elaborate on the connection between the variables. Overall, if you study personal resources and well-being as a main variable, potential theoretical models related to resources and well-being should be considered.
Response 2: Ryff’s model of well-being promotion has been included as an overarching theoretical framework to integrate the variables under study. Additionally, the introduction has been revised to clarify the conceptual connections between personal resources and well-being (see revised lines 47–79).
Comment 3: Reliability coefficients in this sample can be reported in Table 1 as these data are part of results.
Response 3: The reliability coefficients have been removed from the main text and incorporated into Table 1 as suggested.
Comment 4: Please indicate examples of statements used in each subscale in the instruments.
Response 4: Examples of representative items for each subscale have been added in the instruments section for greater clarity.
Comment 5: When describing statistical analyses, I would suggest be more specific and indicate for instance the type of correlation (e.g., Pearson, Spearman etc.). This information is not provided.
Response 5: Pearson correlation coefficients were calculated; this has now been specified in the data analysis section.
Comment 6: Please format tables for better readability. Please decrease the font size.
Response 6: Tables have been reformatted with reduced font size and clearer labeling to enhance readability.
Comment 7: Fit indices should be described in statistical analysis section. There are issues with the abbreviation use. Please make sure that abbreviations are deciphered when introducing.
Response 7: Fit indices have been described in the data analysis section, and all abbreviations have been defined upon first use for clarity.
Comment 8: P-values cannot equal 0.000, in such cases use p < 0.001.
Response 8: P-values have been corrected accordingly (see Table 3).
Comment 9: It seems that covariance between self-esteem and resilience was introduced (Figure 1). Please confirm and elaborate on this if it is the case.
Response 9: Covariance between self-esteem and resilience was indeed introduced in the model, with a value of .61. This relationship is further elaborated in the discussion in connection with the ego-resilience construct (see lines 113 -118 / 362–368).
Comment 10: Please elaborate on potential discriminant validity issues of the studied variables. Theoretical and empirical distinction of these constructs should be demonstrated, especially between self-esteem and well-being.
Response 10: We address potential jingle and jangle fallacies by clarifying that, although constructs like self-esteem and certain dimensions of psychological well-being (e.g., self-acceptance, autonomy) may appear similar, their theoretical distinctions are well-established in the literature and described in the introduction. Our analyses were conducted using total scale scores, but we acknowledge that item-level analyses could further enhance empirical discriminant validity. Future studies may adopt such approaches to explore latent structures. Despite this, we consider our current approach sufficient, given the use of theoretically grounded constructs and psychometrically validated instruments in both original and present samples.

Reviewer 2 Report
Comments and Suggestions for Authors
This paper explores the relationship between emotional competencies and various dimensions of psychological well-being among university students aged 18 to 30 in rural areas of Costa Rica. It also examines the mediating roles of self-esteem and resilience. The methodology is well-chosen and supported by a solid body of literature. By focusing on a vulnerable Latin American population, the study not only contributes to academic understanding but also extends the scope of current research in meaningful ways. I recommend acceptance with minor revisions.
- The literature review is thoughtfully structured and clearly demonstrates that the author has systematically gathered and synthesized relevant studies. As a result, readers are able to gain a solid foundation in the topic before delving into the research.
- The study was approved by the National University of Costa Rica Ethics Committee, indicating that it meets the necessary ethical standards, including informed consent and data protection for all participants.
- In Part 4, the discussion is particularly strong. The author compares current findings with existing literature, creating a thoughtful dialogue that helps readers understand both the relevance and uniqueness of the study's contributions.
- Part 5 offers a concise and focused summary of the study’s key findings. However, it would benefit from a brief reflection on the research challenges encountered, such as the limitation of using only a survey method. While surveys are efficient and cost-effective, they can sometimes lack contextual depth or the ability to capture complex, unstructured responses. Adding this kind of reflection and suggestions for future research would help round out the conclusion.
Author Response
Comment 1: Part 5 offers a concise and focused summary of the study’s key findings. However, it would benefit from a brief reflection on the research challenges encountered, such as the limitation of using only a survey method. While surveys are efficient and cost-effective, they can sometimes lack contextual depth or the ability to capture complex, unstructured responses. Adding this kind of reflection and suggestions for future research would help round out the conclusion.
Response 1: The conclusion has been expanded to include a reflection on methodological limitations, particularly the exclusive use of survey methods, and recommendations for future research directions (see lines 385–392).

Round 2
Reviewer 1 Report
Comments and Suggestions for Authors
The paper has been improved in a satisfactory way.